# Coming out under fire: The role of minority stress and emotion regulation in sexual orientation disclosure

Ilana Seager van Dyk[1,2]*, Amelia Aldao[2,3], John E. Pachankis[1]

**1** Department of Social and Behavioral Sciences, Yale School of Public Health, New Haven, Connecticut, United States of America, **2** Department of Psychology, The Ohio State University, Columbus, Ohio, United States of America, **3** Together CBT, New York, New York, United States of America

* ilana.seagervandyk@yale.edu

**Data Availability Statement:** All data except for the written reflection qualitative data are available on OSF (DOI: 10.17605/OSF.IO/7MR9Y). The written reflection qualitative data cannot be shared publicly because participants cannot be fully deidentified in

## Abstract

Minority stress is hypothesized to interfere with sexual orientation disclosure and sexual minority wellbeing. In this study, we investigated whether minority stress is causally linked to reduced disclosure in sexual minorities, and whether emotion regulation, a potentially adaptive form of stigma coping, can intervene to promote disclosure even following exposure to minority stress. Sexual minority adults in the US ($N = 168$) were recruited online and randomized to a 2 x 2 between-subjects experimental design, where they: 1) received either emotion regulation instructions that asked them to either distance themselves from an emotionally evocative film clip or immerse themselves in the clip, and then 2) viewed either an affirming or a minority stress film clip. Following the film clip, participants completed a written reflection task in which they reflected on the film clip they viewed, which allowed research assistants to subsequently code for participants' spontaneous disclosures of sexual orientation. Participants who viewed the minority stress clip were significantly less likely to spontaneously disclose their sexual orientation in the written task compared to those who viewed the affirming film clip, $OR = 3.21$, 95% CI [1.14, 9.05], $p = .03$. Although the emotion regulation manipulation was successful, there was no effect on sexual orientation disclosure. To our knowledge, this is the first study to demonstrate a causal link between minority stress and disclosure in sexual minorities, and thus highlights an important mechanism underlying minority stress's effects on sexual minority wellbeing. Results demonstrate the importance of interventions that affirm marginalized identities and promote safe sexual orientation disclosure. Future research is needed to determine the circumstances under which effective emotion regulation can buffer against the negative emotional effects of minority stress to promote healthy approach behaviors like disclosure in safe contexts.

## Introduction

Despite recent increases in their acceptance in several countries around the world [1, 2], sexual minorities continue to experience high rates of minority stress (i.e., unique stress related to

the content of many responses, and given the sensitive nature of the sample (i.e., a marginalized community), we wish to minimize the potential of harm to participants. However, all quantitative variables (including those related to the reflections) are included in files on OSF. Data are available from the Ohio State University Office of Responsible Research Practices (ORRP) (contact via hsconcerns@osu.edu) for researchers who meet the criteria for access to confidential data.

**Funding:** The author(s) received no specific funding for this work.

**Competing interests:** The authors have declared that no competing interests exist.

their minoritized identity that heterosexuals do not experience; e.g., sexual orientation-related discrimination, victimization) [3–6]. Minority stress, and expectations thereof, can impair sexual minorities' physical and mental health by potentially hindering sexual orientation disclosure [7, 8]. Disclosure, or "coming out" as a sexual minority, represents a critical component of adaptive minority identity development, and allows individuals access to sexual minority communities through which they can further explore their emerging identity [9]. However, because of pervasive discrimination, many sexual minorities may not feel safe disclosing their sexual orientation to family, friends, colleagues, and healthcare providers, particularly in the absence of explicit safety cues [10–12]. Indeed, concealment of identity-relevant information (as well as negative responses to such disclosure) is associated with cognitive, affective, and behavioral challenges, including hyper-vigilance, shame, and interpersonal avoidance [13, 14], although it is important to acknowledge that these constructs are closely intertwined and behavioral and affective inhibition are also thought to generate concealment [15].

Because sexual orientation disclosure in a safe context has the potential to offer sexual minorities significant social and emotional benefits, understanding potentially modifiable environmental determinants of disclosure represents an important public health concern. For example, understanding the impact of discriminatory press coverage of sexual minorities might enable news organizations to modify their reporting guidelines in order to minimize harm to this vulnerable population. Numerous survey-based studies have shown that sexual minorities who live in high-stigma environments (e.g., places with homophobic laws and policies) are less likely to disclose their sexual orientation [16, 17], and in so doing, they may protect themselves from experiencing even more minority stress (e.g., interpersonal rejection or violence) than would be expected in a high stigma environment. Indeed, some studies find a positive relationship between sexual orientation disclosure and minority stress [18], such that sexual minorities who disclose their sexual orientation are more likely to experience discrimination.

Importantly, none of these studies have utilized experimental methods to demonstrate a causal link between minority stress and concealment. This is an important gap in the literature that precludes causal inference, in that self-reports of concealment might be conflated with self-reports of minority stress through same-source reporting bias or a tendency for people who are more likely to conceal to also be more likely to perceive and report minority stress— possibilities that current non-experimental research cannot rule out. Consequently, this study's primary goal was to elucidate previous correlational findings by experimentally investigating whether minority stress has a direct and immediate effect on disclosure decisions among sexual minority adults, and if so, in which direction. Importantly, we sought to examine the impact of minority stress on day-to-day decisions to disclose one's sexual orientation— for example, choosing whether to disclose that your "wife" is actually a husband when a work colleague invites said wife to a company event. Our rationale is that while "major" disclosures (e.g., to parents, siblings) have been shown to have a substantial impact on mental health [19], there is less research on how these more frequent, arguably "smaller" disclosures are impacted by minority stress.

A secondary goal was to identify personal coping strategies that might buffer against the association between minority stress and concealment. Emotion regulation refers to the processes by which individuals modulate characteristics of their emotions (e.g., type, timing, expression) [20]. Because minority stress is associated with negative mood reactivity [21], as well as negative emotions like shame and sadness [22–24], which are in turn associated with social withdrawal [25], being able to effectively down-regulate one's negative emotions after leaving a discriminatory context (i.e., when the individual is in a safer situation) may increase the likelihood of disclosure in the service of seeking social support to cope with the minority

stress experience. As such, emotion regulation may represent an important coping strategy to consider when examining the impact of minority stress on sexual orientation disclosure [26]. Importantly, given the very real threat to sexual minorities' physical safety in many instances of minority stress (e.g., discrimination, victimization), it is likely adaptive for these individuals to socially withdraw and to conceal their identities while the discriminatory experience is ongoing. However, once the environment is more affirming, it may be useful to down-regulate the emotional and behavioral sequalae of the discriminatory experience in order to increase social connection and coping.

Some emotion regulation strategies are likely more helpful than others in the context of minority stress. For instance, one of the few experimental studies in this area compared the utility of two strategies, rumination and distraction, in response to an idiographic discrimination induction [27]. While discrimination increased self-reported psychological distress across participants, individuals instructed to distract themselves from their distress showed greater reductions in self-reported negative affect compared to those instructed to ruminate. In line with this finding, work by Kross and colleagues demonstrated that individuals who are better able to distance themselves from their emotions and thoughts tend to experience less intense emotional reactions and ruminate less, as compared to individuals who immerse themselves in their emotional experiences [28–30]. Indeed, cross-sectional research has found that sexual minorities who use cognitive change strategies, like distancing, might be protected against the adverse behavioral outcomes of discrimination [31]. Distancing refers to the process of mentally removing oneself from a given situation and is associated with shorter emotion duration and reduced emotional reactivity compared to strategies like immersion (i.e., the process of fully engaging oneself in a given situation [30]). Yet, to our knowledge, no studies have examined the effect of experimentally induced emotion regulation strategies on behavioral responses to sexual orientation-related minority stress. Such research can help inform whether certain emotional regulation strategies might have a socially adaptive benefit in the face of minority stress.

In the present study, we predicted that participants who watched a minority stress film would be less likely, and take longer, to spontaneously disclose their sexual orientation in a written reflection task than those who watched an affirming film. To examine the potentially protective role of emotion regulation, we also predicted that this main effect of film would be qualified by a two-way interaction with emotion regulation, such that among those who viewed the minority stress film, distancing would be associated with more disclosure than immersion.

## Materials and methods

### Participants

Lesbian, gay, and bisexual adults (LGB; *N* = 217) were recruited via targeted postings in social media, Craigslist, listservs, and emails to special interest groups. Postings indicated that the study was about "sexuality and emotions," and were accompanied by rainbow-related images. Sample size was determined based on previous studies examining similar constructs in sexual minorities (scaled up for number of between-subjects conditions [27, 32]), as well as high expected rates of data loss due to the online nature of the study (i.e., incomplete surveys, repeat completions, ineligible participants). Eligible participants needed to: 1) be over the age of 18, 2) self-identify as a sexual minority (i.e., lesbian, gay, or bisexual), and 3) reside in the United States (due to U.S.-centric film stimuli). All study participation was completed online via Qualtrics. Given that online data collection affords less experimental control, we took numerous measures to ensure maximal data integrity. We removed 21 participants for incomplete task

data (e.g., did not write reflection), 13 participants for responses consistent with inattentive responding (i.e., all answers given were "1" and/or incorrect responses to attention questions), three participants for problematic open-response reflections (e.g., off-topic, low-effort), and 12 participants for failing to follow emotion regulation instructions (operationalized as answering 1 "*not at all*" in response to a question about whether they used their assigned strategy while watching the film clip).

The final sample ($n$ = 168) had a mean age of 24.6 years ($SD$ = 7.3 years). A total of 82.1% identified as White; 26.2% identified as men, 57.1% as women, 11.3% as genderqueer, and 5.4% as having another gender identity; and 43.5% identified as bisexual (69.9% female), 26.8% as lesbian, 17.9% as gay, and 11.9% as having another non-heterosexual orientation. Participants were relatively well-educated, with 41.6% indicating that they had obtained at least a four-year college degree. Furthermore, they came from a diverse range of locations, with 42.9% living in a large city, 30.4% living in a small city, 15.5% living in a suburban area, 9.5% living in a small town, and 1.8% living in a rural area, and. See Table 1 for additional demographic characteristics.

## Procedure

All study procedures were approved by The Ohio State University Institutional Review Board under protocol # 2014B0539. After providing informed consent electronically via Qualtrics (documentation of written consent was waived by the IRB given the online nature of the study), participants completed a battery of questionnaires, including demographic questions (see S1 Appendix). We then randomized participants to a 2 x 2 between-subjects design, where they: 1) received either immersion or distancing emotion regulation instructions and 2) viewed either a minority stress or an affirming film clip. Next, participants completed attention questions related to the film clip, and then completed a written reflection task designed to capture spontaneous sexual orientation self-disclosure. At the end of the study, participants were debriefed, linked to an LGB-affirming film clip, and paid $15 for their time (~60 minutes).

## Emotion regulation manipulation

Participants randomly received either immersion ($n$ = 82)—fully engaging with the evocative stimulus—or distancing ($n$ = 86)—mentally removing oneself from an emotionally evocative stimulus—instructions directly prior to their assigned film clip. Immersion instructions were adapted from prior experimental paradigms on perspective-taking [33, 34]:

> As you watch this video, try to <u>embrace the thoughts and feelings you are having</u>. Concentrate on the way <u>you</u> would feel if you were experiencing the events in this video, or if the people in this video were talking to you. Think about how the things you are seeing and hearing relate to <u>who you are as a person</u>, and how you would feel during the rest of your day if you had experienced these events. Imagine as clearly and vividly as possible everything that you would experience—the thoughts, the feelings, everything.

Distancing instructions followed the same format and length:

> As you watch this video, try to be <u>as mentally removed as possible</u> about the events and people you see. Remind yourself that you are not being spoken to by the people in the video. Remind yourself that the opinions and events you are seeing and hearing have <u>no bearing</u> on who you are as a person, on your relationships and friendships, or on how you view

**Table 1. Demographic characteristics of the sample.**

| Demographic characteristic | *n* | % |
|---|---|---|
| Sex assigned at birth | | |
| Female | 124 | 73.8 |
| Male | 44 | 26.2 |
| Gender identity | | |
| • Woman | 96 | 57.1 |
| • Man | 44 | 26.2 |
| • Genderqueer | 19 | 11.3 |
| • Other | 9 | 5.4 |
| Sexual orientation | | |
| • Lesbian | 45 | 26.8 |
| • Gay | 30 | 17.9 |
| • Bisexual | 73 | 43.5 |
| • Other | 20 | 11.9 |
| Race | | |
| • White/Caucasian | 138 | 82.1 |
| • Asian | 13 | 7.7 |
| • Black/African American | 1 | 0.6 |
| • American Indian/Alaska Native/First Nations | 5 | 3.0 |
| • Native Hawaiian/Other Pacific Islander | 2 | 1.2 |
| • Other | 9 | 5.4 |
| Ethnicity | | |
| • Hispanic or Latino | 23 | 13.7 |
| Highest education attained | | |
| • Some high school | 5 | 3.0 |
| • High school diploma/GED | 18 | 10.7 |
| • Some college | 66 | 39.3 |
| • 2-year degree/certificate | 9 | 5.4 |
| • 4-year college degree | 33 | 19.6 |
| • Some post-graduate/professional school | 14 | 8.3 |
| • Post-graduate/professional degree | 23 | 13.7 |
| Occupation | | |
| • Student | 79 | 47.0 |
| • Employed full-time (> = 35 hours/week) | 47 | 28.0 |
| • Employed part-time (<35 hours/week) | 27 | 16.1 |
| • Unemployed | 11 | 6.5 |
| • Homemaker | 4 | 2.4 |
| Region | | |
| • Large city | 72 | 42.9 |
| • Small city | 51 | 30.4 |
| • Suburban area | 26 | 15.5 |
| • Small town | 16 | 9.5 |
| • Rural area | 3 | 1.8 |

*N* = 168. Participants were on average 24.6 years old (*SD* = 7.3; range: 18–66). "Other" gender identities included agender (*n* = 2), demigirl (*n* = 1), demigirl/non-binary (*n* = 1), non-binary (*n* = 3), queer (*n* = 1), and Two-Spirit/genderqueer (*n* = 1). "Other" sexual orientations included asexual (*n* = 1), asexual/biromantic (*n* = 1), homoflexible/queer (*n* = 1), pansexual (*n* = 8), and queer (*n* = 9).

yourself. Whatever emotions you are experiencing right now will come and go—they are only <u>temporary</u> and will have no impact on how you feel for the rest of the day.

Following the film, participants rated their compliance with their assigned emotion regulation instructions on 1 "not at all" to 9 "completely" scale. To assess immersion, participants were asked, "To what extent did you think about the events in the video as though they were happening to you?" To assess distancing, participants were asked "To what extent did you think about the events in the video as though they were happening to someone else?" All participants answered both questions, regardless of emotion regulation condition.

## Film clips

Participants viewed either a minority stress ($n$ = 83) or an LGB-affirming ($n$ = 85) film clip. The discriminatory film clip has been validated in several LGB samples as a reliable induction of both negative affect and sexual orientation-related minority stress [35]. Briefly, it is a two-minute clip consisting of a montage of videos (e.g., TV shows, movies, news programs) depicting LGB people in a negative light in the media, at school and/or work, and at home (e.g., homophobic comments by political commentators, LGB parents being refused service at a restaurant, familial rejection of LGB child). The affirming film clip, which was found online [36], was the same length and consisted of positive depictions of LGB people in the aforementioned settings (e.g., marriage equality victory, familial support of LGB children, proud LGB individuals). Montages were used, rather than using longer clips containing single subjects, in order to broaden the appeal of the clips to different genders and sexual orientations, as well as to increase the likelihood of each participant personally identifying with some part of the film clip. The montage approach also helped ensure adequate stimulus sampling [37]. Prior research on the minority stress clip demonstrates that it elicits self-relevant thoughts about five different minority stress domains in LGB adults who view it [35]; thus, we posit that the clip is a valid induction of sexual orientation-related minority stress.

## Written reflection task

Participants completed a four-minute written reflection task in which they reflected on their thoughts and feelings about the film they watched. We adapted instructions for this task from a previous study investigating differences in self-disclosure between face-to-face interactions and those executed more remotely (e.g., through the internet) [38]. We also asked participants to write about how they did or did not identify with the film. In order to minimize demand characteristics, we asked participants distracter questions (e.g., "What parts of the video (if any) made you feel physical sensations in your body?"). To minimize priming, this task occurred roughly 30 minutes after participants completed the demographic question about sexual orientation. Participants were presented with all question prompts at the top of the page, and then were asked to respond to them in one continuous essay. The full instructions were:

Try to be as honest as possible and describe exactly what you were feeling and thinking as you watched the video. As you write, we would like you to really let go and explore your very deepest emotions and thoughts. You may write about how you identified with different parts of the video, or about similar experiences in your life. Everyone responds differently to this video—we want to know what this video means to you. For example, you might consider the following: What positive or negative feelings (if any) did you experience while watching this video? What parts of the video (if any) shocked or surprised you? Which of your thoughts about the video stand out most to you? What parts of the video (if any) made

you feel physical sensations in your body? If at any time you draw a blank, or run out of things to write, just relax and give yourself time to think about something else related to the topic. Remember: there are no right or wrong answers in this task—all we ask is that you try your best and write from the heart.

To determine whether participants spontaneously disclosed their sexual orientation during the reflection task, two undergraduate research assistants coded each written reflection. Research assistants were blind to film and emotion regulation condition, and were not involved in study design or analyses. A reflection was identified as including disclosure if the participant included their sexual orientation in the text (e.g., "As a lesbian. . ." or "I am very proud to be part of the LGBT community"). Research assistants also coded the number of words until disclosure (calculated as the minimum number of words in the reflection needed to determine the participant's sexual orientation). Discrepancies were resolved by the first author during a consensus meeting.

This paradigm was chosen over a forced choice disclosure paradigm (e.g., asking participants whether they would disclose their sexual orientation to a friend or family member after viewing the film clip) in order to maximize ecological validity. As no other experimental studies of sexual orientation disclosure exist, we developed this approach as an analogue of a sexual minority individual's spontaneous behavior after being exposed to discriminatory media content on social media or television (which is where the film clips were drawn from).

## Attention questions

Attention questions were used to catch automated responding [39, 40]. Participants completed two film-specific multiple-choice questions that focused on the visual (e.g., "One man in the video talked about building fences. Where was he located during this clip?") and auditory (e.g., "In the clip where the son comes out to his mother in the kitchen, what does the mother say in response?") content of the clips in order to ensure that participants were fully engaged with the task. Nine participants from the original 217 were excluded for incorrect responses to attention questions.

## Statistical analyses

All analyses were conducted using SPSS version 26 [41], and the threshold for significance (i.e., α) was set at 0.05. As our data cleaning procedure removed participants who did not complete the experimental task (see Participants), we did not have missing data. All continuous data variables met assumptions of normality. First, as a manipulation check, we ran one-way ANOVAs predicting reported emotion regulation use during the film from the assigned emotion regulation condition. Next, we used logistic regression to examine main and interactive effects of film (minority stress vs. affirming) and emotion regulation (immersion vs. distancing) conditions on sexual orientation disclosure during the reflection task (i.e., dichotomous outcome variable). Post-hoc sensitivity analyses in $G^*Power$ [42] using Hsieh et al.'s procedure [43] indicated the obtained sample provided 80% power to detect effects as small as OR = 0.23 with an $\alpha$-level of .05 in a logistic regression of this kind. Finally, we ran a negative binomial regression predicting number of words until disclosure from emotion regulation and film conditions.

## Results

### Manipulation check

We found a significant difference between the immersion and distancing conditions in the degree to which participants thought about events depicted in the film as though they were

**Table 2. Descriptive characteristics of disclosure variables by film and emotion regulation condition.**

| | Minority Stress | | Affirming | |
| --- | --- | --- | --- | --- |
| | Distancing | Immersion | Distancing | Immersion |
| *n* | 44 | 39 | 42 | 43 |
| Disclosure (%) | 50.0 | 61.5 | 71.4 | 83.7 |
| Average total word count | 142.50 (58.91) | 147.62 (59.97) | 136.95 (64.19) | 142.93 (69.93) |
| Average word count until disclosure | 72.59 (54.57) | 66.33 (40.82) | 58.00 (51.58) | 51.75 (41.53) |

Mean (Standard deviation). Average word count until disclosure includes only participants who disclosed their sexual orientation during the reflection task.

happening to *themselves* (i.e., immersion), $F(1, 166) = 25.97$, $p < .001$, Hedges' $g = 0.79$, such that those in the immersion condition reported greater immersion than those in the distancing condition. We also found a significant difference between the immersion and distancing conditions in the degree to which participants thought about events depicted in the film as though they were happening to *someone else* (i.e., distancing), $F(1, 166) = 10.57$, $p = .001$, Hedges' $g = 0.50$, such that those in the distancing condition reported greater distancing than those in the immersion condition.

## Likelihood of disclosure

We found a significant main effect of film, $OR = 3.21$, 95% CI [1.14, 9.05], $p = .027$, such that participants in the minority stress condition were 3.2 times less likely to disclose their sexual orientation during the reflection task than those in the affirming condition. Contrary to expectations, we found no effect of emotion regulation condition, $OR = 0.63$, 95% CI [0.26, 1.50], $p = .29$, and the interaction was non-significant, $OR = 0.78$, 95% CI [0.20, 3.05], $p = .72$. See Table 2 for descriptive characteristics. Although we were underpowered to statistically examine differences by sexual orientation group (lesbian/gay, bisexual, other), descriptive characteristics show that much fewer bisexual participants (34.8%) disclosed in the minority stress/distancing conditions than lesbian/gay participants (76.5%), suggesting that there may be differences by sexual orientation in how participants responded to these manipulations (see S1 Table).

## Words to disclosure

Although descriptive results (see Table 2) were in line with hypotheses, there was no significant main effect of film condition, $OR = 1.25$, 95% CI [0.82, 1.92], $p = .30$, or emotion regulation, $OR = 0.89$, 95% CI [0.61, 1.30], $p = .55$, on number of words until disclosure and the interaction was non-significant, $OR = 1.02$, 95% CI [0.57, 1.84], $p = .94$. See S1 Table for descriptive characteristics by sexual orientation group.

## Discussion

This study sought to examine whether 1) minority stress has a causal effect on sexual orientation concealment, and 2) emotion regulation can buffer against negative effects of minority stress on disclosure. Indeed, participants exposed to minority stress content were less likely to spontaneously disclose their orientation than those who viewed affirming content. This experimental finding clarifies the correlational findings found in previous research by identifying a causal direction from minority stress to disclosure. This finding also supports the notion that sexual minorities exposed to environments in which minority stress is common might be more likely to conceal their sexual orientation. In fact, previous research has shown

associations between discriminatory national laws, policies, and community attitudes and reduced odds of sexual orientation disclosure, with negative implications for health [16, 17, 44, 45]. Given that the minority stress content in this study consisted of video clips from commonly watched television and online media sources, our findings also point to the importance of developing and following media guidelines for accurate, inclusive, and affirming reporting on sexual minorities and their experiences in order to minimize harm [46]. As societies make progress toward these types of structural and institutional reductions in discrimination, clinical health interventions with sexual minorities can address the role of minority stress on disclosure and related outcomes. Indeed, psychosocial interventions that affirm marginalized identities and help sexual minorities to cope with minority stress can promote not only greater sexual orientation disclosure (in safe contexts) [47], but also ultimately influence broader societal change by empowering sexual minority individuals to be the agents of that change [48]. Interventions that promote disclosure specifically have been argued to perpetuate civil rights gains of this historically marginalized population [49, 50].

We did not find evidence for our hypothesis that emotion regulation would buffer against minority stress and facilitate disclosure. Although successful manipulation checks suggest that participants understood and tried to follow the emotion regulation instructions, perhaps the manipulation (i.e., immersion vs. distancing) was not strong enough to alter participants' emotional responses. Future studies should employ different emotion regulation strategies (e.g., cognitive reappraisal, suppression), as well as a no-instructions control, to determine whether a different strategy can buffer the impact of minority stress on disclosure behavior. It could also be interesting to combine this affective science approach with Miller and Kaiser's theoretical model of stigma coping [51]. For example, a future study could compare primary control coping strategies like regulating emotion expression with secondary control coping strategies like cognitive restructuring. Another possible explanation for these results could be that the instructions for the written reflection task overrode the emotion regulation instructions for the distancing group. Although participants in the distancing condition may have followed regulation instructions while watching the film clip (as indicated by the manipulation check), they likely also followed the written reflection task instructions (delivered a couple minutes later), which directed participants to explore their "very deepest emotions and thoughts." Thus, participants may have been more likely to disclose than if they had been encouraged to continue distancing themselves from their emotions. Future research should consider changing the instructions for the written reflection task (e.g., by removing any language that may conflict with the emotion regulation instructions) to determine if doing so reveals an effect of emotion regulation on disclosure.

We also did not find support for our predictions related to the timing of sexual orientation disclosure in the written reflection task. While descriptive statistics (Table 2) were in line with our hypotheses (i.e., the minority stress clip was associated with more words until disclosure than the affirming clip; the distancing condition was associated with more words until disclosure than the immersion condition), it is likely that a larger sample size is needed in order to reach statistical significance.

Our study provides initial insight into the frequency of sexual orientation disclosure in the context of minority stress and affirmation. Given our findings, future research could take a closer look at the depth, breadth, and duration of participants' disclosure to see whether these qualities predict wellbeing over time. A large body of literature has shown that self-disclosure about personal information can play an important role in facilitating intimacy and relationship development (for a review, see [52]); however, less is known about how these features of disclosure play out in stigmatized populations and for other outcomes, including mental and physical health.

Future studies should also expand their minority stress analogues to include not only film clips, but also interactive paradigms (e.g., online chat interface, in-person interactions). Given that many of the minority stress experiences LGB individuals face in their lives occur interpersonally (e.g., in schools and workplaces; [3, 53]), it is critical that future work captures this additional dimension in order to better understand the real-life experiences of LGB individuals and thus their full affective repertoire during times of adversity. In addition to interpersonal minority stress experiences, recent work has shown that structural stigma (i.e., characteristics of a social environment that oppress LGB individuals such as policies excluding sexual minorities from institutions like marriage) may have a significant impact on the lives and wellbeing of LGB individuals [54]. As such, it is crucial that future investigations consider the role of structural stigma and the affective and biological processes related to it.

Although this study has a number of strengths, including its use of innovative experimental methods and the ecological validity of the film clips, there are also several limitations that should be addressed in future studies. First, participants who were recruited knew that the study was about sexuality, which may have influenced their responses (e.g., participants may have felt more comfortable disclosing their sexual orientation if they assumed the research team were LGBTQ-affirming). Given this limitation, it is a testament to the strength of the effect of minority stress that although participants had already disclosed their orientation to the research team, they then chose, consciously or unconsciously, to withhold the information after viewing the minority stress clip. Nonetheless, the paradigm should be repeated with naïve samples with a wide range of sexual minority identities (including questioning individuals) to confirm the results. Similarly, this study was underpowered to examine differences among individuals with different sexual minority identities. Since descriptive results seemed to suggest differences in the effects of the manipulations on lesbian/gay vs. bisexual participants, it is crucial that future studies with larger samples of diverse sexual identities investigate these possibilities. Next, it is possible that participants did not identify with the film clip they viewed, and thus did not write about their sexual orientation in relation to the clip (rather than avoiding disclosing their sexual orientation as a result of the minority stress content). While the film clips purposefully included a montage of smaller clips in order to minimize this possibility, future work should consider matching minority stress analogues to participants' experience (e.g., using an idiographic approach involving prescreening minority stress experiences and selecting a relevant film clip). Relatedly, due to power constraints, this study did not examine the potential moderating effects of individual differences (e.g., prior sexual orientation disclosure, internalized stigma) on the relationship between minority stress exposure and disclosure. Future studies should recruit larger samples with a high degree of variation in these individual difference variables in order to determine whether and how these factors influence the strength of the minority stress-disclosure relationship. This study also did not assess participants' explicit motivations for disclosing their sexual orientation during the reflection task. As such, it is unclear whether participants would respond the same way in a real-life conversation, or if the decision to disclose was related to some feature of the study paradigm. Additional studies with different minority stress analogues and more diverse samples of LGB individuals are needed to determine the generalizability of this finding. Finally, due to the difficulty of creating an ecologically valid stimulus that is "neutral" regarding sexual minority individuals, this study did not employ a neutral control condition that would allow for conclusions to be made about whether results reflect a reduction in spontaneous disclosure after minority stress exposure versus an increase in disclosure after affirmation. Future research should consider including such a control so that this can be assessed.

This is the first known study to experimentally demonstrate a causal link between minority stress and sexual orientation disclosure. Although additional research is needed to examine

the parameters under which this effect operates, this finding highlights an important mechanism potentially underlying minority stress's effects on sexual minority identity development and ultimate wellbeing. These findings can support public policy and psychosocial interventions to advance the public visibility of this historically marginalized population, whose members face frequent pressures to conceal an important aspect of their identity. In so doing, these findings may help reduce the mental health burden that sexual minority individuals disproportionately experience.

## Supporting information

**S1 Appendix. Contents of questionnaire battery.**
(DOCX)

**S1 Table. Descriptive characteristics of disclosure variables by film, emotion regulation condition, and sexual orientation.**
(DOCX)

## Acknowledgments

The authors would like to thank Jennifer Cheavens, Kara Christensen, Lisa Cravens-Brown, and Jennifer Crocker for their feedback on previous versions of this manuscript. They would also like to thank Thomas Parsons and Sarah Gobrial for their aid during data collection.

## Author Contributions

**Conceptualization:** Ilana Seager van Dyk, Amelia Aldao.

**Data curation:** Ilana Seager van Dyk, Amelia Aldao.

**Formal analysis:** Ilana Seager van Dyk, Amelia Aldao, John E. Pachankis.

**Funding acquisition:** Ilana Seager van Dyk, Amelia Aldao.

**Investigation:** Ilana Seager van Dyk, Amelia Aldao.

**Methodology:** Ilana Seager van Dyk, Amelia Aldao, John E. Pachankis.

**Project administration:** Ilana Seager van Dyk.

**Supervision:** Amelia Aldao, John E. Pachankis.

**Writing – original draft:** Ilana Seager van Dyk, Amelia Aldao, John E. Pachankis.

**Writing – review & editing:** Ilana Seager van Dyk, John E. Pachankis.

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
