## [Decision Letter · Decision Letter 0]

7 Feb 2022

PONE-D-21-20910Coming out under fire: The role of discrimination and emotion regulation in sexual orientation disclosurePLOS ONE

Dear Dr. Seager van Dyk,

Thank you for submitting your manuscript to PLOS ONE. After careful consideration, we feel that it has merit but does not fully meet PLOS ONE’s publication criteria as it currently stands. Therefore, we invite you to submit a revised version of the manuscript that addresses the points raised during the review process.

We have completed the editorial process for your manuscript. The topic of this manuscript is incredibly important and you’ve developed a novel method for examining the causal effects of what you term “discrimination” on spontaneous disclosure. Two expert reviewers have also read and commented on the manuscript. While all of the reviewers comments were positive, there are a few areas that I believe can be strengthened to be considered for publication. Accordingly, the editorial recommendation is to revise and resubmit for further review. Please note that resubmitting your manuscript does not guarantee eventual acceptance, and that your resubmission will be subject to re-review by the Action Editor before a decision is rendered.

I’ll start first with summarizing and adding to the reviewers’ comments and then contribute my own comments below.

Reviewer 1 astutely asked why you didn’t include a neutral clip about sexual minorities (and potentially heterosexual couples) with facts, images, and figures that was neither positively or negatively valenced. This would allow you to make claims about whether disclosure increases or decreases compared to these neutral clips. While material is rarely perceived as neutral and may have been why you didn't include this type of "control" condition, it would be nice to know whether disclosure increases or decreases compared to this control condition rather than that there is a difference between affirming and discriminatory stimuli (two extreme cases). I think it is a good question for you to consider and pose for future work. While I know this type of study takes time and resources in terms of data collection, cost, and coding of the spontaneous disclosure, I wonder whether you could conduct a Study 2 with this type of clip. Reviewer 1 also brings up whether you’re powered to detect an interaction in your model. Please address this question in your method or discussion.

Reviewer 2 brings up a very important point about the clarity of the language you use to describe the video manipulations. I agree that what you use as discrimination/discriminatory is really exposure to discriminatory or disconfirming stimuli. I liked the idea that Reviewer 2 posed that this could potentially be a minority stress induction used in the lab. It had me thinking about the Trier Social Stress Test and how that’s a stress induction in the lab that is commonly used. I would like you to carefully reconsider what you call your manipulation with an eye towards being as clear and true to what your stimuli actually presented and what scenario/experience your participants were in when they were assigned to one condition vs. the other. This is not an issue with just this study, but an issue with the field at large in being clear on language so that studies in the future can be compared against each other as this is not the same as discrimination experiences that one would complete on a survey. It is as if you're priming people in a way and I would urge you to look into other literature that manipulates these types of identity cues (e.g., Cipollina, R., & Sanchez, D. T. (2021). Identity cues influence sexual minorities’ anticipated treatment and disclosure intentions in healthcare settings: Exploring a multiple pathway model. *Journal of health psychology*, 1359105321995984.)

Finally, Reviewer 2 has comments about how your distancing manipulation may be at odds with exploring inner emotions/thoughts. Please comment on this in your revision letter and in the manuscript.

My comments follow:

On p3, first paragraph you introduce how pervasive discrimination may impact disclosure to various groups and leave out healthcare providers- one area of work that has received much attention and where disclosure is important (with training by providers on how to respond and appropriate care). In this same paragraph you lay out a very causal path by which concealment induces processes and feelings (hypervigilance and shame) yet it could go the other way as well. Can you dampen the language in the causal paths here and acknowledge these processes are intricately tied. It’s actually a strength of your research, is that you’re trying to disentangle these effects. Which brings me to another question- did you measure any individual differences that impacted the strength of your manipulations (i.e. moderators) on disclosure? For example, ability to perspective take or outness, internalized homophobia etc…

I appreciated that you acknowledged your sample was already out enough to join a study on LGBTQ issues. I wonder if you included questioning individuals what your effects would look like.

On p4 last paragraph, you start discussing the non-adaptive effects of emotion regulation when experiencing discrimination, however, if I read this right, I would disagree. I think it’s very adaptive for individuals who experience discrimination in their environment to conceal and withdraw because their environment is not safe. Can you explain what you mean here if I’ve misinterpreted and clarify for your reader and also balance this view. You only briefly note this perspective middle of p5.

On p4, line 82-83, I don’t understand this sentence, “being able to effectively down-regulate one’s negative emotions in discriminatory contexts may increase the likelihood of disclosure, as a form of prosocial behavior.”….what does this have to prosocial behavior/helping behavior?

While I appreciate your sample size is small, can you present results on whether disclosure differed by sexual minority identity?

How long, on average, did the study last? Please include this when you say participants were paid $15 for their time.

For transparency, please include what the battery of questionnaires included.

P9 line 170-173, please clarify that both groups answered the two questions. I didn’t understand this until your results.

How many participants were excluded based on your “attention questions”, you list excluded earlier but not here.

Please interpret your odds ratios for the reader.

While I know this is not a qualitative paper, can you give a few examples of how people disclosed?

We look forward to receiving your revised manuscript.

Kind regards,

Mollie A Ruben, Ph.D.

Academic Editor

PLOS ONE

Journal Requirements:

2. Please note that in order to use the direct billing option the corresponding author must be affiliated with the chosen institute. Please either amend your manuscript to change the affiliation or corresponding author, or email us at plosone@plos.org with a request to remove this option.

5. We note that you have referenced (ie. Bewick et al. [5]) which has currently not yet been accepted for publication. Please remove this from your References and amend this to state in the body of your manuscript: (ie “Bewick et al. [Unpublished]”) as detailed online in our guide for authors

Reviewers' comments:

Reviewer's Responses to Questions

**Comments to the Author**

1. Is the manuscript technically sound, and do the data support the conclusions?

Reviewer #1: Yes

Reviewer #2: Yes

2. Has the statistical analysis been performed appropriately and rigorously? 

Reviewer #1: Yes

Reviewer #2: Yes

3. Have the authors made all data underlying the findings in their manuscript fully available?

Reviewer #1: Yes

Reviewer #2: No

4. Is the manuscript presented in an intelligible fashion and written in standard English?

Reviewer #1: Yes

Reviewer #2: Yes

5. Review Comments to the Author

Reviewer #1: This was a well-written manuscript that demonstrated greater spontaneous disclosure of sexual minority status (coming out) after group affirming rather than discriminatory filmclips. It would have been interesting to include a neutral film clip (perhaps a montage of facts and figures about sexual minorities in the US that is neither affirming nor discriminatory) to examine whether results reflect a reduction in disclosure after discrimination versus an increase in spontaneous disclosure after affirmation - and perhaps the authors could speculate about this in the discussion. Moreover, I wonder if N of 168 would be sufficient power to get a 2X2 interaction (the original hypothesis re: emotion regulation) on a measure like spontaneous disclosure. In general though, this was a well done study with an interesting ecologically valid measure of disclosure.

Reviewer #2: This manuscript details the results of an online experimental study assessing the impact of viewing a sexual-minority-affirming or sexual-minority-discriminatory film clip, as well as the impact of receiving emotion regulation instruction (either immersion or distancing), on the likelihood of spontaneously disclosing a sexual minority orientation. The research is well designed and the experimental approach provides novel insights. Thus, the manuscript represents an important contribution to the literature. I have two minor concerns; I believe that if the authors address these concerns, it will strengthen the manuscript.

First, the study is framed as an experimental discrimination induction design. “Discrimination” is not defined in the literature review and the reader is left to use the common usage understanding of discrimination, which is more interpersonal in nature (e.g., another person commits a discriminatory act toward someone because of their sexual orientation). However, the study design includes exposure to discriminatory stimuli. Rewording the literature review and discussion sections to replace “discrimination” with “exposure to discriminatory stimuli” would be more precise, but would be excessively wordy and make the manuscript difficult to read. However, a broader discussion in the literature review of what constitutes discrimination, including specific mention of exposure to discriminatory stimuli such as the video clip used in the current study, would help to make the paper clearer. There is discussion of how the video clip is a useful minority stress induction tool in the methods section; however, similar information needs to be included in the literature review section to more fully frame the construct of discrimination being studied in the present research.

Second, the authors note in the discussion section that the emotion regulation intervention did not impact disclosure. However, the distancing instructions speak to being mentally removed but the written reflection instructions speak of exploring “very deepest emotions and thoughts;” these appear to be at odds with one another. Given that the timing between the emotion regulation instructions and the written reflection instructions appears to only be a couple minutes, it is quite possible that the reflection instructions cancelled out any emotion regulation induction for those in the distancing group. This seems to be an important limitation that should be explored in the discussion.

6. PLOS authors have the option to publish the peer review history of their article (what does this mean?). If published, this will include your full peer review and any attached files.

Reviewer #1: No

Reviewer #2: **Yes: **Nathan Grant Smith

---

## [Author Response · Author response to Decision Letter 0]

15 Mar 2022

Please see attached file for better formatted responses.

Response to Reviewers

March 14, 2022

RE: PONE-D-21-20910

Dear Dr. Ruben and reviewers,

Thank you for the opportunity to revise our manuscript entitled “Coming out under fire: The role of minority stress and emotion regulation in sexual orientation disclosure.” We are grateful for the thoughtful feedback provided by both the editor and the reviewers, and we believe that the suggested adjustments have substantially strengthened the manuscript. In response to your comments, we now: (1) clarify our language when discussing the film clip manipulations by reframing the “discriminatory” stimulus as a “minority stress” stimulus; (2) include a post-hoc sensitivity analysis to address reviewer concerns about power; (3) provide additional details in the method regarding the length of the study, use of attention questions, how disclosure was operationalized; (4) address the lack of a control condition in this experiment and discuss the difficulties associated with designing this kind of “neutral” stimulus in research about sexual minorities; and (5) review several limitations in the discussion section including the potential conflict between the emotion regulation instructions and the written reflection instructions. In this document, we highlight the reviewers’ comments in italics and then provide our detailed responses below. All edits to the manuscript have been highlighted in yellow for ease of review.

Thank you again for your valuable feedback and your time.

Editor comments:

Comment 1: Reviewer 1 astutely asked why you didn’t include a neutral clip about sexual minorities (and potentially heterosexual couples) with facts, images, and figures that was neither positively or negatively valenced. This would allow you to make claims about whether disclosure increases or decreases compared to these neutral clips. While material is rarely perceived as neutral and may have been why you didn't include this type of "control" condition, it would be nice to know whether disclosure increases or decreases compared to this control condition rather than that there is a difference between affirming and discriminatory stimuli (two extreme cases). I think it is a good question for you to consider and pose for future work. While I know this type of study takes time and resources in terms of data collection, cost, and coding of the spontaneous disclosure, I wonder whether you could conduct a Study 2 with this type of clip. 

Response: Thank you for this important suggestion. We agree that a future study that includes some kind of “neutral” control clip would be informative. As you anticipated, we elected not to use a neutral control condition because of the difficulty of creating such a clip, as well as our focus on using ecologically valid stimuli that participants might encounter in their daily lives. Although the reviewer suggests that we include perhaps a montage of facts and figures about sexual minorities in the US that is neither affirming nor discriminatory, in the current climate it is unclear what facts and figures about sexual minorities could be included in such a clip that would be considered neutral. Statistics about the number of individuals identifying as sexual minorities could be viewed as affirming (i.e., a sign that a sexual minority is not alone), as could simply depicting same-sex couples, given the continued lack of representation of LGB people in some areas of the media. Facts about the fairly unchangeable nature of sexual orientation have been vociferously debated by anti-LGBTQ groups like the Family Research Council who promote so-called “conversion” practices — thus inclusion of such facts could be considered affirming and not neutral. Facts about the impact of discrimination and stigma on sexual minorities has also been debated by these same anti-LGBTQ groups, so any stimuli related to this area (e.g., news clips) could be considered non-neutral. Even definitional information (such as the notion that bisexuality means an individual is attracted to more than one gender) could be considered not neutral, by virtue of affirming sexual minorities’ rights to self-identify. Indeed, the Family Research Council encourages its members to “avoid using the terms “gay,” “lesbian,” or “bisexual” as solo nouns because this tends to imply that some people’s intrinsic, inborn, immutable identity as gay, lesbian, etc. is who they are… We instead say “people who engage in homosexual conduct” or “people who identify as homosexual.”” Thus, even using identity labels could be considered affirming. Importantly, since sexual orientation is a concealable identity, it is necessary for identity labels to be used in the video stimuli. Otherwise, it would be unclear that the stimulus is supposed to be related to sexual orientation. Of note, both discriminatory and affirming clips included videos from news broadcasts (affirming: about marriage equality; discriminatory: about banning LGBTQ content in schools), which could be considered more “neutral” stimuli, in that news broadcasts could be considered more objective than for example, the more “extreme” views expressed in online commentary and editorials. A note regarding the need for additional research that includes a more neutral control condition has been added to the limitations section on page 20, lines 402-407.

Family Research Council, How to Respond to the LGBT Movement. URL: https://www.frc.org/get.cfm?i=BC18B01

Comment 2: Reviewer 1 also brings up whether you’re powered to detect an interaction in your model. Please address this question in your method or discussion.

Response: We have added a post-hoc sensitivity analysis addressing this concern to the statistical analysis section on page 13, lines 263-265. 

Post-hoc sensitivity analyses in G*Power (42) using Hsieh et al.’s procedure (43) indicated the obtained sample provided 80% power to detect effects as small as OR = 0.23 with an �-level of .05 in a logistic regression of this kind. 

Comment 3: Reviewer 2 brings up a very important point about the clarity of the language you use to describe the video manipulations. I agree that what you use as discrimination/discriminatory is really exposure to discriminatory or disconfirming stimuli. I liked the idea that Reviewer 2 posed that this could potentially be a minority stress induction used in the lab. It had me thinking about the Trier Social Stress Test and how that’s a stress induction in the lab that is commonly used. I would like you to carefully reconsider what you call your manipulation with an eye towards being as clear and true to what your stimuli actually presented and what scenario/experience your participants were in when they were assigned to one condition vs. the other. This is not an issue with just this study, but an issue with the field at large in being clear on language so that studies in the future can be compared against each other as this is not the same as discrimination experiences that one would complete on a survey. It is as if you’re priming people in a way and I would urge you to look into other literature that manipulates these types of identity cues (e.g., Cipollina, R., & Sanchez, D. T. (2021). Identity cues influence sexual minorities’ anticipated treatment and disclosure intentions in healthcare settings: Exploring a multiple pathway model. Journal of health psychology, 1359105321995984.)

Response: Thank you for this important suggestion. We agree that language is important, especially when operationalizing constructs in experimental studies. Since the discriminatory film clip has been validated as a minority stress induction more broadly (see citation 35 in the manuscript), we have decided to instead refer to this clip throughout the manuscript (and in the title) as the “minority stress” clip. We think this is a more precise description of this manipulation, since the clip validation study found that participants who view this film clip describe thinking about multiple different types of minority stress (including discrimination, violence, internalized homophobia, concealment, expectations of rejection), not just discrimination. Thank you also for drawing our attention to the Cipollina & Sanchez study — we have added reference to it on page 3.

Comment 4: Finally, Reviewer 2 has comments about how your distancing manipulation may be at odds with exploring inner emotions/thoughts. Please comment on this in your revision letter and in the manuscript.

Response: Thank you for highlighting this potential conflict between the instructions for the emotion regulation manipulation, and the instructions for the written reflection task. We agree that this may explain our emotion regulation results, and have added the following text to the discussion on page 17, lines 336-346.

Another possible explanation for these results could be that the instructions for the written reflection task overrode the emotion regulation instructions for the distancing group. Although participants in the distancing condition may have followed regulation instructions while watching the film clip (as indicated by the manipulation check), they likely also followed the written reflection task instructions (delivered a couple minutes later), which directed participants to explore their “very deepest emotions and thoughts.” Thus, participants may have been more likely to disclose than if they had been encouraged to continue distancing themselves from their emotions. Future research should consider changing the instructions for the written reflection task (e.g., by removing any language that may conflict with the emotion regulation instructions) to determine if doing so reveals an effect of emotion regulation on disclosure.

Comment 5: On p3, first paragraph you introduce how pervasive discrimination may impact disclosure to various groups and leave out healthcare providers- one area of work that has received much attention and where disclosure is important (with training by providers on how to respond and appropriate care). In this same paragraph you lay out a very causal path by which concealment induces processes and feelings (hypervigilance and shame) yet it could go the other way as well. Can you dampen the language in the causal paths here and acknowledge these processes are intricately tied. It’s actually a strength of your research, is that you’re trying to disentangle these effects. 

Response: This is an excellent point. We have amended this paragraph to include healthcare providers, as well as to acknowledge the complex relationships between concealment and other cognitive, behavioral, and affective processes. This paragraph now reads as follows:

However, because of pervasive discrimination, many sexual minorities may not feel safe disclosing their sexual orientation to family, friends, colleagues, and healthcare providers, particularly in the absence of explicit safety cues (10–12). Indeed, concealment of identity-relevant information (as well as negative responses to such disclosure) is associated with cognitive, affective, and behavioral challenges, including hyper-vigilance, shame, and interpersonal avoidance (13,14), although it is important to acknowledge that these constructs are closely intertwined and behavioral and affective inhibition are also thought to generate concealment (15).

Comment 6: Which brings me to another question- did you measure any individual differences that impacted the strength of your manipulations (i.e. moderators) on disclosure? For example, ability to perspective take or outness, internalized homophobia etc…

Response: Given the primary aim of this study (to examine main and interactive effects of film clip and emotion regulation on disclosure), as well as concerns about being underpowered to detect moderation effects in addition to our predicted effects, we do not include moderation analyses in this manuscript. While we did collect questionnaires related to outness and internalized stigma as well as other relevant sexual minority identity variables (see S1 Appendix), due to the sheer number of relevant variables, we do not wish to inadvertently conduct a fishing expedition by running and reporting all possible results. However, we do think that this is an important direction for future research, and we have added the following comment to this effect in the discussion on page 19, lines 392-396.

Relatedly, due to power constraints, this study did not examine the potential moderating effects of individual differences (e.g., prior sexual orientation disclosure, internalized stigma) on the relationship between minority stress exposure and disclosure. Future studies should recruit larger samples with a high degree of variation in these individual difference variables in order to determine whether and how these factors influence the strength of the minority stress-disclosure relationship.

Comment 7: I appreciated that you acknowledged your sample was already out enough to join a study on LGBTQ issues. I wonder if you included questioning individuals what your effects would look like.

Response: We think this would be an interesting future study as well! Since our operationalization of disclosure (as described on page 12, lines 235-236) required participants to in some way indicate their non-heterosexual sexual orientation in their written reflection task response (e.g., “As a gay man…” or “When I came out…”), we imagine it may be challenging to capture this kind of casual disclosure in quite the same way with questioning individuals (beyond participants responding “As someone who is questioning their sexual orientation…”). To our knowledge, there are few studies that specifically focus on questioning individuals, likely due to recruitment challenges. Hollander (2000) also posits that questioning youth may be less likely to explicitly disclose their questioning status for two main reasons:

1. In line with several developmental models of sexual identity development, questioning youth have yet to find an identity label that fits their experience, so do not have words to describe their experience when asked, and 

2. Anti-LGBTQ stigma and discrimination likely prompt questioning youth to assume a heterosexual identity when asked to disclose in order to avoid maltreatment.

We have added a brief note to the discussion about the need for more research on this topic (see page 19, line 381).

Comment 8: On p4 last paragraph, you start discussing the non-adaptive effects of emotion regulation when experiencing discrimination, however, if I read this right, I would disagree. I think it’s very adaptive for individuals who experience discrimination in their environment to conceal and withdraw because their environment is not safe. Can you explain what you mean here if I’ve misinterpreted and clarify for your reader and also balance this view. You only briefly note this perspective middle of p5.

Response: Thank you for highlighting the need for additional clarity in this section. We have revised this section to reflect the importance of considering the adaptiveness of concealment and avoidance in situations where disclosing one’s sexual minority status could increase safety concerns. Specifically, we rephrased as follows (see page 5, lines 85-97):

Because minority stress is associated with negative mood reactivity (21), as well as negative emotions like shame and sadness (22–24), which are in turn associated with social withdrawal (25), being able to effectively down-regulate one’s negative emotions after leaving a discriminatory context (i.e., when the individual is in a safer situation) may increase the likelihood of disclosure in the service of seeking social support to cope with the minority stress experience. As such, emotion regulation ¬may represent an important coping strategy to consider when examining the impact of minority stress on sexual orientation disclosure (26). Importantly, given the very real threat to sexual minorities’ physical safety in many instances of minority stress (e.g., discrimination, victimization), it is likely adaptive for these individuals to socially withdraw and to conceal their identities while the discriminatory experience is ongoing. However, once the environment is more affirming, it may be useful to down-regulate the emotional and behavioral sequalae of the discriminatory experience in order to increase social connection and coping.

Comment 9: On p4, line 82-83, I don’t understand this sentence, “being able to effectively down-regulate one’s negative emotions in discriminatory contexts may increase the likelihood of disclosure, as a form of prosocial behavior.”….what does this have to prosocial behavior/helping behavior?

Response: Thanks for this catching this error. In line with the previous comment, we have rephrased this paragraph to be clearer.

Comment 10: While I appreciate your sample size is small, can you present results on whether disclosure differed by sexual minority identity?

Response: Thank you for this suggestion. While our small sample size and subsequent low power precludes adding a condition x sexual orientation group interaction term to our models, we have added a supplemental table (S1 Table) that includes the descriptive characteristics of our outcome variables broken down by sexual orientation group (lesbian/gay, bisexual, and other), as well as by film and emotion regulation conditions. We included a brief description of the most notable observation in the results section on page 14-15:

Although we were underpowered to statistically examine differences by sexual orientation group (lesbian/gay, bisexual, other), descriptive characteristics show that much fewer bisexual participants (34.8%) disclosed in the minority stress/distancing conditions than lesbian/gay participants (76.5%), suggesting that there may be differences by sexual orientation in how participants responded to these manipulations (see S1 Table).

We also noted the importance of investigating differences in disclosure by sexual orientation in future studies in the discussion on page 19, lines 382-386:

Similarly, this study was underpowered to examine differences among individuals with different sexual minority identities. Since descriptive results seemed to suggest differences in the effects of the manipulations on lesbian/gay vs. bisexual participants, it is crucial that future studies with larger samples of diverse sexual identities investigate these possibilities.

Comment 11: How long, on average, did the study last? Please include this when you say participants were paid $15 for their time.

Response: The study took participants around 60 minutes to complete. This detail was added to page 9, line 166.

Comment 12: For transparency, please include what the battery of questionnaires included.

Response: We have added the list of questionnaires included in the study to S1 Appendix. 

Comment 13: P9 line 170-173, please clarify that both groups answered the two questions. I didn’t understand this until your results.

Response: Thank you for this suggestion. We have clarified that all participants answered both questions, regardless of emotion regulation condition on page 10, lines 189-190.

Comment 14: How many participants were excluded based on your “attention questions”, you list excluded earlier but not here.

Response: Nine participants were excluded based on incorrect responses to attention questions. We have added this information to page 7, line 136, and page 13, lines 252-253.

Comment 15: Please interpret your odds ratios for the reader.

Response: Thank you for this suggestion. We have made this adjustment to the results section. 

Comment 16: While I know this is not a qualitative paper, can you give a few examples of how people disclosed?

Response: We have included examples of how participants disclosed in our description of the coding process on page 12, lines 235-236. 

Reviewers' comments:

Comment 17: Reviewer #1: This was a well-written manuscript that demonstrated greater spontaneous disclosure of sexual minority status (coming out) after group affirming rather than discriminatory film clips. It would have been interesting to include a neutral film clip (perhaps a montage of facts and figures about sexual minorities in the US that is neither affirming nor discriminatory) to examine whether results reflect a reduction in disclosure after discrimination versus an increase in spontaneous disclosure after affirmation - and perhaps the authors could speculate about this in the discussion. 

Response: Thank you for raising these important points. As described in our response to Comment 1, we agree that a future study that includes some kind of “neutral” control clip would be informative. We elected not to use a neutral control condition because of the difficulty of creating such a clip, as well as our focus on using ecologically valid stimuli that participants might encounter in their daily lives. As described in more detail in our response to Comment 1, it is unclear what facts and figures about sexual minorities could be included in such a clip that would be considered neutral, and indeed even definitional information (such as the notion that bisexuality means an individual is attracted to more than one gender) could be considered not neutral, by virtue of affirming sexual minorities’ rights to self-identify. Of note, both discriminatory and affirming clips included videos from news broadcasts (affirming: about marriage equality; discriminatory: about banning LGBTQ content in schools), which could be considered more “neutral” stimuli, in that news broadcasts could be considered more objective than for example, the more “extreme” views expressed in online commentary and editorials. A note regarding the need for additional research that includes a more neutral control condition has been added to the limitations section on page 20, lines 402-407.

Comment 18: Moreover, I wonder if N of 168 would be sufficient power to get a 2X2 interaction (the original hypothesis re: emotion regulation) on a measure like spontaneous disclosure. In general though, this was a well done study with an interesting ecologically valid measure of disclosure.

Response: Thank you for your kind words about our study. As noted in response to Comment 2, we have added a post-hoc sensitivity analysis addressing this concern to the statistical analysis section on page 13, lines 263-265. 

Post-hoc sensitivity analyses in G*Power (42) using Hsieh et al.’s procedure (43) indicated the obtained sample provided 80% power to detect effects as small as OR = 0.23 with an �-level of .05 in a logistic regression of this kind. 

Comment 19: Reviewer #2: This manuscript details the results of an online experimental study assessing the impact of viewing a sexual-minority-affirming or sexual-minority-discriminatory film clip, as well as the impact of receiving emotion regulation instruction (either immersion or distancing), on the likelihood of spontaneously disclosing a sexual minority orientation. The research is well designed and the experimental approach provides novel insights. Thus, the manuscript represents an important contribution to the literature. I have two minor concerns; I believe that if the authors address these concerns, it will strengthen the manuscript.

First, the study is framed as an experimental discrimination induction design. “Discrimination” is not defined in the literature review and the reader is left to use the common usage understanding of discrimination, which is more interpersonal in nature (e.g., another person commits a discriminatory act toward someone because of their sexual orientation). However, the study design includes exposure to discriminatory stimuli. Rewording the literature review and discussion sections to replace “discrimination” with “exposure to discriminatory stimuli” would be more precise, but would be excessively wordy and make the manuscript difficult to read. However, a broader discussion in the literature review of what constitutes discrimination, including specific mention of exposure to discriminatory stimuli such as the video clip used in the current study, would help to make the paper clearer. There is discussion of how the video clip is a useful minority stress induction tool in the methods section; however, similar information needs to be included in the literature review section to more fully frame the construct of discrimination being studied in the present research.

Response: Thank you for your kind words about our study, and for these useful suggestions. After reflecting on your comment, we decided to change our language throughout the manuscript by referring to the discriminatory film clip as the “minority stress” clip. We think this is a more precise description of this manipulation, since a study validating this film clip as a minority stress induction (citation 35 in the manuscript) found that participants who view this film clip describe thinking about multiple different types of minority stress (including discrimination, violence, internalized homophobia, concealment, expectations of rejection), not just discrimination.

Comment 20: Second, the authors note in the discussion section that the emotion regulation intervention did not impact disclosure. However, the distancing instructions speak to being mentally removed but the written reflection instructions speak of exploring “very deepest emotions and thoughts;” these appear to be at odds with one another. Given that the timing between the emotion regulation instructions and the written reflection instructions appears to only be a couple minutes, it is quite possible that the reflection instructions cancelled out any emotion regulation induction for those in the distancing group. This seems to be an important limitation that should be explored in the discussion.

Response: Thank you for highlighting this potential conflict between the instructions for the emotion regulation manipulation, and the instructions for the written reflection task. We agree that this may explain our emotion regulation results, and have added the following text to the discussion on page 17, lines 336-346.

Another possible explanation for these results could be that the instructions for the written reflection task overrode the emotion regulation instructions for the distancing group. Although participants in the distancing condition may have followed regulation instructions while watching the film clip (as indicated by the manipulation check), they likely also followed the written reflection task instructions (delivered a couple minutes later), which directed participants to explore their “very deepest emotions and thoughts.” Thus, participants may have been more likely to disclose than if they had been encouraged to continue distancing themselves from their emotions. Future research should consider changing the instructions for the written reflection task (e.g., by removing any language that may conflict with the emotion regulation instructions) to determine if doing so reveals an effect of emotion regulation on disclosure.

---

## [Editor Report · Decision Letter 1]

18 Apr 2022

Coming out under fire: The role of minority stress and emotion regulation in sexual orientation disclosure

PONE-D-21-20910R1

Dear Dr. Seager van Dyk,

We’re pleased to inform you that your manuscript has been judged scientifically suitable for publication and will be formally accepted for publication once it meets all outstanding technical requirements.

Kind regards,

Mollie A Ruben, Ph.D.

Academic Editor

PLOS ONE

Additional Editor Comments (optional):

I think this is a wonderful contribution to the field. Thank you for doing responding to all of the editor's and reviewers' comments so thoughtfully and doing such important rigorous research on these topics. It was a real pleasure reading the manuscript the first time and even more so now with the clarity of the revisions. Best, Mollie

---

## [Editor Report · Acceptance letter]

22 Apr 2022

PONE-D-21-20910R1 

Coming out under fire: The role of minority stress and emotion regulation in sexual orientation disclosure 

Dear Dr. Seager van Dyk:

I'm pleased to inform you that your manuscript has been deemed suitable for publication in PLOS ONE. Congratulations! Your manuscript is now with our production department. 

Kind regards, 

on behalf of

Dr. Mollie A Ruben 

Academic Editor

PLOS ONE